# Design and validation of a new Healthcare Systems Usability Scale (HSUS) for clinical decision support systems: a mixed-methods approach

Abir Ghorayeb ![ORCID],[1] Julie L Darbyshire ![ORCID],[2] Marta W Wronikowska ![ORCID],[1] Peter J Watkinson ![ORCID] [3]

[1]Nuffield Department of Clinical Neurosciences, University of Oxford, Oxford, UK
[2]Department of Primary Care Health Sciences, Oxford University, Oxford, UK
[3]Kadoorie Centre for Critical Care Research and Education, Oxford University Hospitals NHS Trust, Oxford, UK

**Correspondence to**
Dr Abir Ghorayeb;
abir.ghorayeb@bristol.ac.uk

## ABSTRACT

**Objective** To develop and validate a questionnaire to assess the usability of clinical decision support systems (CDSS) and to assist in the early identification of usability issues that may impact patient safety and quality of care.

**Design** Mixed research methods were used to develop and validate the questionnaire. The qualitative study involved scale item development, content and face validity. Pilot testing established construct validity using factor analysis and facilitated estimates for reliability and internal consistency using the Cronbach's alpha coefficient.

**Setting** Two hospitals within a single National Health Service Trust.

**Participants** We recruited a panel of 7 experts in usability and questionnaire writing for health purposes to test content validity; 10 participants to assess face validity and 78 participants for the pilot testing. To be eligible for this last phase, participants needed to be health professionals with at least 3 months experience using the local hospital electronic patient record system.

**Results** Feedback from the face and content validity phases contributed to the development and improvement of scale items. The final Healthcare Systems Usability Scale (HSUS) proved quick to complete, easy to understand and was mostly worded by potential users. Exploratory analysis revealed four factors related to patient safety, task execution, alerts or recommendations accuracy, the effects of the system on workflow and ease of system use. These separate into four subscales: patient safety and decision effectiveness (seven items), workflow integration (six items), work effectiveness (five items) and user control (four items). These factors affect the quality of care and clinician's ability to make informed and timely decisions when using CDSS. The HSUS has a very good reliability with global Cronbach's alpha 0.914 and between 0.702 and 0.926 for the four subscales.

**Conclusion** The HSUS is a valid and reliable tool for usability testing of CDSS and early identification of usability issues that may cause medical adverse events.

## STRENGTHS AND LIMITATIONS OF THIS STUDY

⇒ Iterative, user-centred approach, designed with potential users.
⇒ Mixed research methods allow comprehensive assessment of the development and evaluation process.
⇒ Panel of experts in usability and questionnaire writing for health purpose review the questionnaire content validity.
⇒ Final scale reliability assessment demonstrates ability of scale to pinpoint usability issues.
⇒ Relatively small number of participants for the pilot study.

intelligently filtered or presented at appropriate times, to enhance patient care'.[1] To benefit from the extensive capabilities of artificial intelligence, health organisations develop CDSS to improve health outcomes for patients and to assist clinicians make appropriate and informed decisions. Ultimately, the intent of any CDSS is to improve clinical efficiency and safety by avoiding adverse events or errors.[2] Clinicians, often working under pressure, are concerned with relevant information access, efficient task completion and better collaboration or communication with colleagues. The design of the CDSS or health information systems (HIS) in general is critical. Poor usability may significantly impact patient care and safety.[3–5] Usable HIS, and more particularly CDSS, would be expected to reduce medical errors, increase efficiency, reduce the cognitive workload of clinicians, and improve patient safety and care. This can be achieved through intuitive design and easily accessible pertinent information that allows users to make timely and informed decisions. Expectations surrounding CDSS development often are too optimistic.

Adoption of CDSS by clinicians is limited due mainly to their poor usability,[6 7] the high

## INTRODUCTION

Clinical decision support systems (CDSS) are defined as systems that 'provide clinicians or patients with computer-generated clinical knowledge and patient-related information,

volume and complexity of data to analyse and the failure to fit new systems within existing clinical workflows.[6 8] Other factors related to system utility, patient safety and impact on professional skills development may also influence the acceptance and use of the CDSS technology.[6 9] Clinicians need to perceive the benefit of patient safety or improved outcomes to be keen to adopt a new CDSS.[9 10] They are prepared to learn a hard to use or complex system if it is perceived as useful and helpful in complex situations.[6] However, poor usability may impact patient safety, lead to frustrated users, workflow disruptions, increase of medical errors and decreased efficiency.[3–5] Consequently, designers must be mindful of the various and varying needs of clinicians, taking into consideration patient safety and quality of care.

## Usability

Usability is defined by the international organisation for standardisation (ISO) as: 'the extent to which a product can be used by specified users to achieve specified goals with effectiveness, efficiency and satisfaction in a specified context of use'.[11 12] It is generally seen as the ability of the technology to allow users to achieve their tasks safely, effectively, efficiently and pleasantly within a specified context or environment.[13 14]

Different usability aspects appear to be critical for healthcare system acceptability. Patient safety, prevention of medical errors, information exchange, collaboration between colleagues and the complexity of navigation through the system can all influence the clinician cognitive workload. Previous research work on usability of HIS and CDSS suggests that poor usability contributes to the decreased cognitive performance of clinicians,[15 16] low efficiency,[17] workflow interruption or disorientation,[6 8 9 18 19] raised medical error risks[20] and higher numbers of adverse events.[21] The electronic health record (EHR), which makes available to clinicians vast amounts of patient data and other information that could be used by CDSS, is increasingly suffering from poor usability and safety issues.[22 23] EHRs are often perceived by users as difficult to use, frustrating and a source of patient safety hazards, and systems that do not integrate into clinical environments and personal workflow well.[24]

Detecting usability issues in the early stages of CDSS development can reduce or highlight human errors and improve patient safety. It is critical to find an adequate way to involve clinicians in the design process from the beginning. However, usability engineering methods are hard to apply and can be time consuming due to the use of various technology systems, complexity of tasks to evaluate, patient and health professionals' recruitment process. They also need to be conducted within a heterogeneous group of healthcare staff with different backgrounds, needs, skills and experiences.[4 25 26] Most of the usability evaluation studies of CDSS to date have used conventional usability methods. Interviews or focus groups,[27] think-aloud protocols,[28] non-validated questionnaires[4 27] or standardised tools like the Systems Usability

Scale (SUS)[28 29] are all commonly used to evaluate CDSS. To the best of our knowledge, specific standards for evaluating the user centred design of CDSS do not exist,[30] and different institutions, organisations or vendors use their own guidelines. For example, Russ et al[31] developed a usability testing method to evaluate health technology software influenced by human factors techniques and a few studies have tried to develop non-validated usability questionnaires to assess the usability of HIS.[4 27 32] There is therefore a critical need to improve usability standardisation practice for CDSS, and health technology systems in general.[22 33]

## Need for a new usability scale for clinical interfaces

Usability is measured by evaluating the interaction between users, tools, tasks and the context of use. Various researchers have developed measurements tools for usability aspects such as efficiency, satisfaction and learnability.[34 35] Questionnaires are the most often applied usability test[36] due to the simplicity of gathering subjective experiences of a large group of participants in a timely and cost-effective way.[37] However, standardised usability questionnaires such as the Questionnaire for User Interface Satisfaction[38] or SUS[29] focus on evaluating the user interface of the system. They do not take into account the wider context of use which is in this case the clinical work.[37] One of the most widely used scale is the SUS, which was originally developed to evaluate the ease of use of systems.[29] The SUS is a standardised questionnaire which consists of 10 statements with which participants specify their agreement or disagreement on a Likert scale. Usually, it is used as a primary method of data collection in interface evaluations but sometimes it is one of several methods of data collection.

Due to the lack of a validated tool which takes into consideration the usability aspects related to the clinical context, SUS was the most commonly used scale[36] to assess users' satisfaction of HIS or CDSS.[9 10 28 33 39 40] In Nair et al,[7] the SUS was applied to evaluate the usability and ease of use of a CDSS for chronic pain management in primary care but identified difficulties in interpretation of the meaning of the score due to the complexity of the clinical task. In the authors' opinion, the SUS neither connects, nor interacts well with the behaviours of the participants observed during the assessment sessions. Another research study[9] used an iterative evaluation of a CDSS through two phases in clinical settings and noticed that the system was rated as usable in the first phase and more usable in the second phase on the SUS while the accompanying qualitative study indicates different degrees of usability in each phase. In phase 1, participants reported the interaction with the system was not intuitive, and that it was time consuming to find the pertinent information. Some of them suffered from a lack of information or additional details needed in the decision-making process. Clinical participants considered the systems disruptive to workflow and felt it took them longer to follow the CDSS' recommendations which in turn limited their use of the

CDSS. These results were congruent with findings from an earlier phase of the Hospital Alerting Via Electronic Noticeboard (HAVEN) project,[41] which confirmed that the SUS score may not reflect improvements in interface performance directly. Therefore, the SUS and existent scales are not reflective of the usability in the real world as it does not consider the context of use (in this case, the clinical context). Furthermore, they do not allow evaluators to identify specific usability problems that can be targeted for improvement.

Researchers have previously tried to address this problem by generating their own questionnaire or survey to evaluate the usability of HIS and have used these questionnaires without a clear or systematic validation.[4 27 32]

A usability scale specifically designed to assess systems in clinical settings can help identify usability problems in an early stage of development or implementation that, if not addressed, may otherwise lead to medical adverse events. To the best of our knowledge, there is no instrument or scale to measure usability of CDSS or HIS that integrates differential usability variance such as patient safety and quality of care. Therefore, there is a critical need for a CDSS/HIS usability scale that takes into consideration different usability aspects, and which is also easy and quick to complete and to analyse. An ideal scale would offer an overview of the usability status of the system with some instructions or information highlighting the specific improvement areas on which to focus.

This scale development work is based on a review of the CDSS and HIS usability literature[42] and on our practical experience from usability testing a CDSS system. This Healthcare SUS (HSUS) was designed within the context of the evaluation of a complex CDSS, HAVEN.[41 43] HAVEN has been designed to be a fully integrated HIS that combines an early warning score with a navigable user interface. Prior to developing the HSUS we evaluated HAVEN for usability, understanding, efficiency and accuracy. We used the results from these assessments to refine the interface design through an iterative process. While we demonstrated good performance indicators for the HAVEN system, we identified a gap in current evaluation methods. Specifically, we found the common methods used to evaluate usability of information systems lacked specificity in the clinical context. The aim of this paper therefore is to present the processes used to develop and validate a usability scale that takes into consideration the clinical context and is suitable to evaluate the usability of any healthcare system.

## METHODS
To develop and validate a new scale, there is a specific methodology in the literature, which combines qualitative and quantitative data effectively.[44–50] These mixed methods have been applied to the development and validation of different scales.[50–55] We used these methods in our study in an exploratory design[56] where we collected

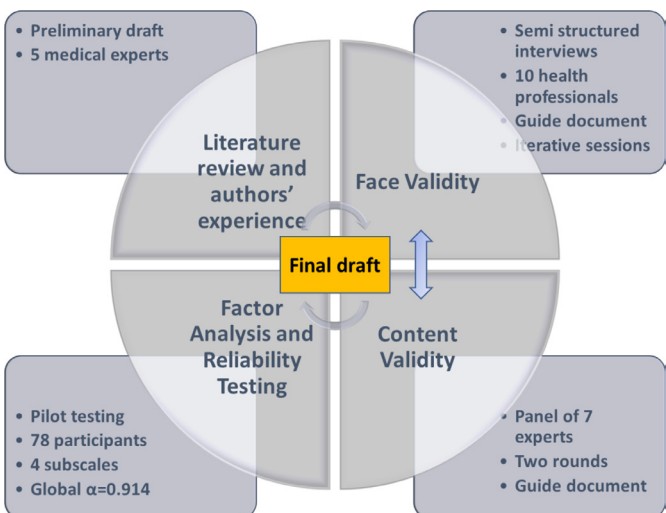

**Figure 1** Phases of development and validation of HSUS scale. HSUS, Healthcare Systems Usability Scale.

qualitative data first and followed this with a quantitative phase. We conclude with a sequential synthesis.

The HSUS development in this study was iterative. A demographic questionnaire was designed to include a variety of users. To develop a valid, reliable and worthwhile usability scale for HIS in general, and CDSS in particular, mixed research methods were used. The main phases in the development and validation of the HSUS are summarised in figure 1.

We proceeded as follows:
a. We established the theoretical background based on a literature review and previous usability evaluation sessions of the HAVEN system and authors' experience. As an output of this stage, we developed the first draft of the questionnaire items.
b. We assessed face validity and content validity of the scale using 10 health professionals and a panel of 7 experts in usability and questionnaire writing for health purpose. These were part of the qualitative research study.
c. We analysed the construct validity of the new scale through pilot testing, factor analysis and internal consistency of the scale. This was the basis for the quantitative research study.

### Patient and public involvement
Two lay members of the public were part of the Hospital Alerting Via Electronic Noticeboard (HAVEN) project management committee and were supportive of the need for a HSUS.

### Content validity
A crucial phase in the validity of a new scale is to ensure that key items have not been missed or omitted.[57 58] The content validity of a questionnaire is usually attained by 'proof reading' by a panel of experts familiar with the construct being measured. Using the panel of experts to review the scale items in depth improves its quality, understandability and representativeness. Using 6–10 experts is

recommended to generate a consensus on the construct of interest.[59]

We invited a consulting committee of experts to review the questionnaire content validity. Experts were chosen based on their knowledge of the usability construct and their experience working in healthcare. The panel included experts in usability assessment of HIS, plain language, design of health content, questionnaire design for usability and health purposes, and clinical informatics. An introduction letter and content validation form were created to define the aim of the scale and explain to the experts what we needed from them. The background questionnaire, the Healthcare System Usability Scale, an introductory letter and the content validation form were sent via email. On completion, all items were returned electronically and comments from the expert panel were either integrated into the questionnaire or captured in a separate document. In this phase, we wanted to understand if the questions captured the usability of the CDSS effectively in the clinical context. The experts were asked to simulate filling out the survey while writing notes and comments, check for common errors (such as double-barrelled questions, double negatives, confusing and leading questions), check questions' clearness and relevance to the main construct, check for any missing aspect of the construct in question, check the question distribution and categorisation, and propose any new aspects not covered by the questionnaire. Finally, they were asked to give each question a score from 1 to 10 which refers to the extent each item relates to the usability of the system.

### Face validity

Face validity refers to the degree to which the purpose of the scale is obvious to the target population or users and seems to be a valid measure for its purpose.[50 60] Face validity evaluates the questionnaire in terms of 'feasibility, readability, consistency of style and formatting and the clarity of the language used'.[61 62] Ideally, the participants who assess questionnaires for face validity will be representative of the target population (the end users).

The participants representing future end users who were approached to assess face validity were recruited from two hospitals within the Oxford University Hospitals National Health Service Foundation Trust (OUHFT), the John Radcliffe Hospital and the Churchill Hospital. We recruited through in-person and email invitation. To be eligible we needed participants to be: (1) Health professionals (physician, nurse, physiotherapist, pharmacist or scientist), (2) Have more than 3 months experience using the local hospital electronic patient record system (SEND or CERNER Millenium) and (3) Willing to participate in the study.

Prior to the start of the study, all participants signed written informed consent. The question items were presented to allow discussion of the usability and healthcare dimensions to be evaluated or to be included in the scale; ensure the inclusion of any usability missing aspect;

review the set of questions and contribute to the usability, understandability and clarity of the question items.

For this project, an assessment form was developed to help the interviewer and participants assess questions in terms of clarity, understandability and meaning, layout and style, and the ability of the participants to answer the questions. Face validity of the HSUS was determined by a review of the items iteratively by 10 participants. Suggestions from each participant were incorporated in the next iteration evaluated.

Each participant's interview was recorded for analysis. After each interview, the participant's feedback was analysed and the questionnaire was amended to be evaluated by the following participants (by adding, deleting or amending a question item). The original questionnaire consisted of 28 multiple choice questions and four open ended questions. This was tested iteratively with 10 health professionals until data saturation was reached.

During the interviews, we used the think aloud technique.[63] This requires the participant to explain out loud each thought they have while reading the questions and discussing the different suggested statements. We were looking to learn how participants speak intuitively about system usability, aspects of patient care and how they interpret statements using their own words. Participants were asked to select the statement they thought best reflected the construct in question, and to change wording, delete statements or add new aspects they thought relevant (figure 2).

### Pilot testing

The last validation step is to pilot test the scale where participants of the target population complete the questionnaire independently. For the validation and the comparator usability testing of the scale, we used two well-known health systems used at the OUHFT: SEND and CERNER Millenium. SEND is a CDSS used by health professionals at the hospital to record patient vital signs observations. This is a fully integrated system developed locally. It allows healthcare staff to record and monitor patient vital signs electronically from the bedside. The system integrates with the Trust electronic patient record system and incorporates a simple data entry interface with a visual trend of historic values to aid clinical decision making in hospital. The SEND system also includes limited advice to support escalation processes in the event that physiological deterioration is identified. We also asked participants to assess CERNER Millenium, which is a different HIS also in use at the OUHFT. We know from previous evaluations that healthcare staff rate SEND significantly more usable than CERNER Millenium. Other work has recognised significant limitations in the visual appearance of the CERNER Millenium interface, which is likely to hinder acceptability and satisfaction in comparison to the simplicity of the SEND interface.[64]

In the piloted questionnaire, we added the option of 'statement not clear' to identify any statement that might need improvement. The option 'not applicable' was also

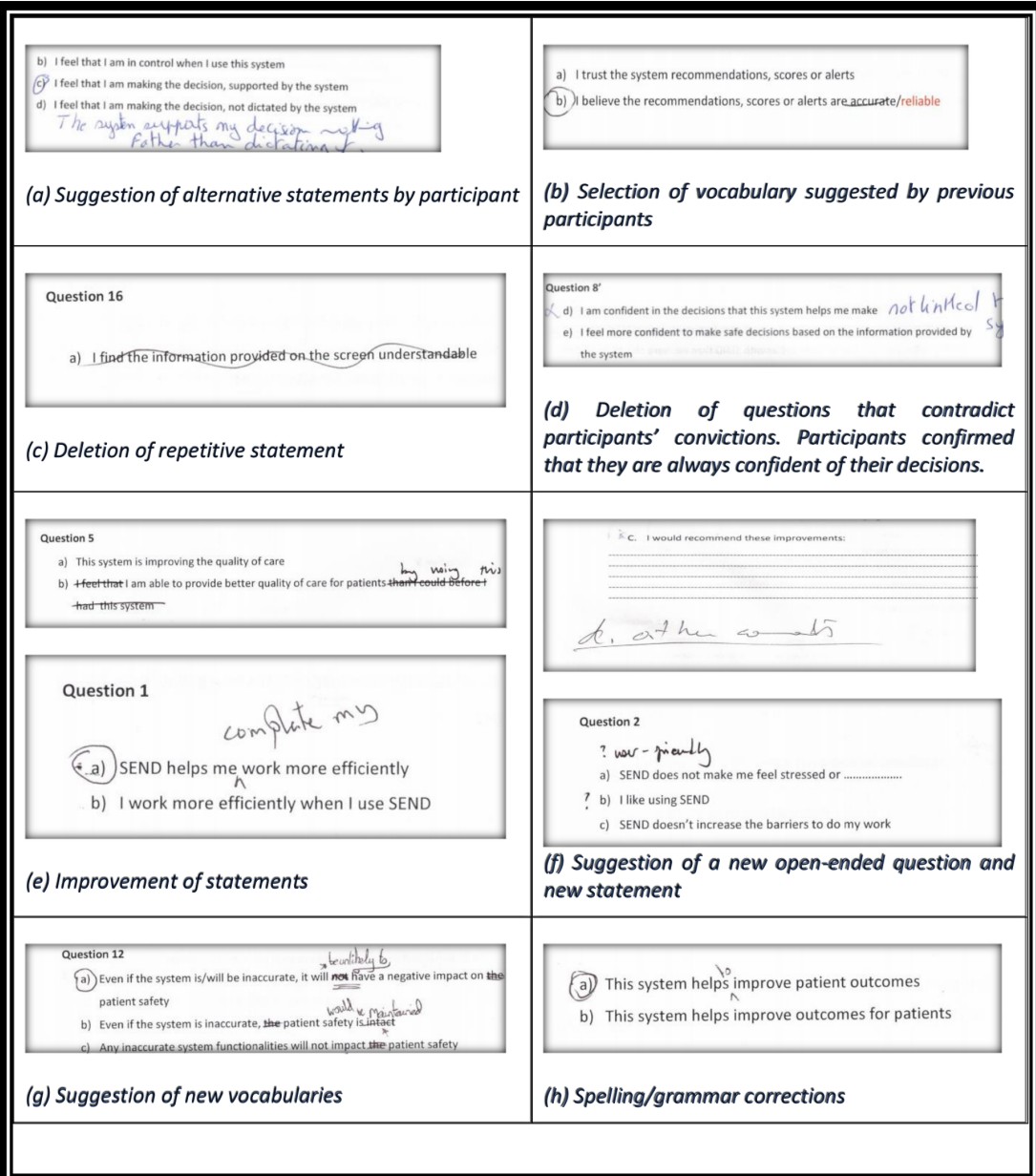

**Figure 2** Selected examples of improvement suggested by the participants.

added as not all HIS or CDSS offer the same functionalities. For example, CERNER Millenium can be used to record and monitor patients' vital signs observations but does not include a visual graphical display to aid interpretation of data. CERNER Millenium also includes direct links to the patient admissions and treatment records which SEND does not.

Participants were recruited from two hospitals within the OUHFT, the John Radcliffe Hospital and the Churchill Hospital through in-person and email contacts. The participants represented a range of experience. Eligibility criteria were the same as for those recruited to assess the HSUS for face validity and consisted of: (1) Health professionals (physician, nurse, physiotherapist, pharmacist or scientist), (2) Have more than 3 months experience using SEND or CERNER Millenium

and (3) Willing to participate in the study. At the beginning of the questionnaire, participants completed and approved a consent form and answered the demographic questions.

The questionnaire was available online and offered as a paper copy. The researchers systematically observed and recorded any difficulties encountered during the process. Data were reviewed to highlight any items that were difficult to understand or generate confusion and to assess the reliability and correlations of the scale. Exploratory factor analyses, which are more convenient for new scales' development,[65] were conducted to identify the underlying constructs or factors. Items measuring the same construct were clustered into the same subscale. Finally, the internal consistency and correlation of the whole scale, and for each subscale, was investigated using

Cronbach's alpha coefficient to ensure scale items were consistent.

## RESULTS

The first draft of the questionnaire was based on the usability definitions from CDSS literature to cover the different usability aspects including learnability, efficiency, effectiveness, usefulness, accessibility and user satisfaction.[66 67] We also wanted to acknowledge the clinical context in which this system is intended to be deployed so included additional patient safety and quality of care aspects. The main challenge to creating an effective scale involves writing questions which are unambiguous and understandable using the vocabulary of the target population. This first draft was developed to cover the main usability issues which mainly linked to:

► Understanding how the recommendations, scores or alerts are made[18]: Clinicians emphasise the need to comprehend how the CDSS' decisions, scores or alerts were calculated or developed. It is part of their training to probe for reasons behind results as interpretations of the outcome or critical issues may differ despite apparently identical data. The 'black box' approach is disliked almost universally.

► Ease of access to additional information: The main aim of all CDSS is to provide an intuitive design to allow pertinent information to be easily accessible to facilitate timely informed decisions. Clinicians must be able to locate the specific information they require promptly.

► Perceived utility: One of the main adoption barriers of CDSS is a lack of perception of its utility. Health professionals need to feel the new technology will help them improve patient outcomes, improve their decision effectiveness and ensure patient safety.

► Cognitive workload: A CDSS that requires significant mental effort to use is most likely to be rejected by health professionals working in a busy environment.

► Workflow integration: The new system must fit into the workflow of the health professionals and integrate well with current systems and processes. Recommendations, alerts or scores should be provided and displayed in a way that supports the clinical work; otherwise, the system is likely to be seen as a waste of time.

To elaborate the first draft of the questionnaire, we collaborated with five scientists working in the medical field to ensure the inclusion of all construct aspects and the appropriateness of each question.

### Content validity

The first set of 21 discreet choice questions and 4 open-ended questions was reviewed by the panel of experts. All experts emphasised the need for a new, more clinically focused usability scale to assess CDSS and HIS, as evidenced by the following quotations:

Expert A: 'I don't know of any survey instrument that focuses specifically on electronic decision support systems. Therefore, developing well-researched questionnaires that can help you distinguish between more usable and less usable systems is worthwhile'.

Expert B: 'I think the questionnaire can be an excellent contribution to the field. I really like the idea of embedding context as that is critical to perceptions about a system.'

Expert C: 'I think each of the subscales is relevant to the concept of usability.' I also think the respondent burden (the time and effort for respondents to fill out the questionnaire) is fine'.

They commented on the questions' clearness, appropriateness and understandability, and looked for leading, confusing or double-barrelled questions and suggested corrections. The panel of experts were asked to rate each item on a scale of 1–10, indicating how closely each item related to the system's usability. We were looking to check whether these items or factors need to be weighted or balanced in some manner. Most experts disagreed with the concept of assigning a score, arguing that it relies on the system being evaluated, its functioning and role. It is up to the researcher or investigator to decide whether certain needs, such as patient safety and decision effectiveness, should be prioritised. Some items are more important than others and data analysis should take this into account. While the HSUS does not weight items, it does allow for the quick and easy identification of the source of usability issues by investigating each subscale separately.

The comments from the expert panel guided us to rethink and reword several of the items. For example, for the first item 'I feel satisfied while using the system', some of the experts suggested to interview potential users with the following: 'This question makes me feel …….' and ask users to suggest an adjective or to choose from a range of adjectives using the Microsoft Product Reaction Cards.[68] We were also using the term 'complexity of tasks' and the experts wondered how respondents would interpret this. What tasks do they think this system is helping them to manage? Or is the complexity coming from something that's nothing to do with the system? The participants in the face validity phase proposed we exclude the term 'complexity of tasks' and replace this with a daily workload assessment. Other advice included separating legibility, findability and understandability as these are distinct needs.

More generally, the panel of experts guided us through our questionnaire development by suggesting corrections, reordering of questions and advising we needed to be more specific and ask only what is necessary for analysis. They suggested additional questions which might be useful for analysis such experience with electronic healthcare systems other than the specific system being evaluated. They found the open-ended items easy to understand, and thought they would be useful to detect aspects of the construct not otherwise well represented or

omitted elsewhere. They also suggested we evaluate the background questionnaire with potential users and ask them for suggestions.

Three of the seven experts offered further review after usability testing of the amended version. This usability testing altered both the content and structure of the questionnaire. Subsequently a revised version of the questionnaire was drafted, and a usability and understandability test was developed during the face validity phase. This amended version of the questionnaire was given to the three experts for a second review and a teleconference was organised to discuss feedback and agree consensus. We made some additional modifications before the pilot testing phase.

### Face validity

The recruited cohort for this phase represented a range of experience, age and skills. The sample consisted of four nurses with different levels of expertise, two physiotherapists, two doctors in training and two consultants. The final scale consists of 22 items rated on a 7-point Likert scale from 'strongly disagree' to 'strongly agree' including 'not applicable' and 'statement not clear'. Data saturation occurred when participants had no more concerns or new suggestions, and we noticed agreement within the last four participants.

We present direct comments and some examples of participants in figure 2. For example, in question 5, participants were asked to define 'quality of care' to confirm consistency of meaning. We were asking the participants a series of questions designed to elicit more information about how framing a question in different ways might affect participants' thoughts, understanding and reasons of selection. Each time a person suggested a new item, new terminology or a statement's correction, the document was changed and reviewed by the next participant. This phase of testing helped us revise some double-barrelled and unclear statements while taking into consideration participants' insights, experience and points of view.

Participants' responses were summarised on a spreadsheet (table 1) for more detailed analysis. The last four participants selected the same statement and shared the same understanding. In table 1, we summarise examples of three statements, selected statements and justification for the selection.

The last version of the amended document was discussed with the three experts who had offered to follow the progress of the work. They advised to proceed with the pilot testing and adopt the suggestions, even though some of these suggestions contradicted the panel of experts' advice. For example, in question 14, many experts suggested to avoid the use of the word 'navigate' as it is classified as computer jargon and suggested the following statement instead: 'It is easy to get to what I need in this system'. However, during the usability sessions users advised the use of the word 'navigate' which might be justified by the daily use of computer technology by all

the participants and health professionals in general. As a result of the second round of interviews with the panel of experts, a final draft questionnaire was ready for the pilot testing.

According to the results of the qualitative studies, most of the items were reworded, four items were deleted and six items were added. The final version of the questionnaire put forward for pilot testing consisted of 23 items.

### Pilot testing

The sample consisted of 78 participants aged 18–65 years. Most of the participants use their local electronic healthcare system daily for at least 1–3 hours, have experience in healthcare for more than 1 year, and work in various healthcare departments as shown in figure 3.

To investigate if the new HSUS was able to differentiate between a system that is perceived to be hard-to-use (CERNER Millenium) and a system perceived to be easy-to-use (SEND), we piloted the questionnaire with 37 healthcare professionals using CERNER Millenium and 41 using SEND. Although the aim was not to evaluate SEND and CERNER Millenium specifically, the results of the survey revealed that the usability score for CERNER Millenium was 64% while the general HSUS score for SEND was 81%, with 86% for the workflow integration subscale.

The cumulative distribution function of the subscale 'patient safety and decision effectiveness' for SEND system (figure 4) is an example of the further investigation offered by the HSUS, and it clearly shows a high usability score, with only a few participants scoring less than 80%.

The HSUS included three optional open-ended questions: 'I would add these functions or features', 'I would change the following', 'Please use the space below to provide any other comments'. Participants' comments were concise, informative, to the point and represent the users' perspectives, as demonstrated in table 2 collected during the pilot testing phase. The differing usability scores for SEND and CERNER Millenium can be understood in context from these user comments.

### Factor analysis

Factor analysis is a statistical method that clusters items into underlying constructs.[69] Each construct is a list of questions or items that can be associated with each other. Authors have argued on exactly how big samples should be as there is no simple rule-of-thumb[70] and it is rarely justified in the literature.[71] Inappropriate sample size might lead to inaccurate findings during the development and validation of a new scale, in particular the identification of the structure of the new scale and its subdivision in latent constructs. The sample can be determined 'by the nature of the data, that is, the stronger the data, the smaller the sample size can be. Strong data display uniformly high communalities without cross-loadings'.[72]

To check the sample size's adequacy to undertake exploratory factor analysis for this study, we performed Bartlett's

**Table 1** Selected examples of items development during face validity

| User no | Question 2 | Question 5 | Question 9 |
|---|---|---|---|
| 01 (senior nurse) | I like using SEND | This system is improving the quality of care | I understand how the system derived its recommendations, scores or alerts |
| 02 (Physiotherapist) | I like using SEND because of its ease of use/because it is stress free | I am able to provide better quality of care for patients than I could before I had this system | I understand how the system derived its recommendations, scores or alerts |
| 03 (pharmacist) | I like using SEND | This system is improving the quality of care | I understand how the system derived its recommendations, scores or alerts |
| 04 (training doctor) | SEND does not make me feel stressed | I feel that I am able to provide better quality of care for patients than I could before I had this system | The system's recommendations are congruent with clinical current practices and standards/conform |
| 05 (consultant) | SEND does not increase the barriers to do my work | I feel that I am able to provide better quality of care for patients than I could before I had this system | The system's recommendations are consistent with clinical practices and standards |
| 06 (senior nurse) | SEND does not make me feel stressed | My colleagues and I are able to provide better quality of care for patients than we could before we had this system | The system's recommendations are consistent with clinical practices and standards |
| 07 (training doctor) | SEND is user-friendly | I am able to provide better quality of care for patients by using this system | 'I understand how the system derived its recommendations, scores or alerts' and 'The system's recommendations are consistent with clinical practices and standards'. (Clinicians should have been using the system for a while to be able to decide this item) |
| 08 (Doctor) | SEND is user-friendly | I am able to provide better quality of care for patients by using this system | 'I understand how the system derived its recommendations, scores or alerts' and 'The system's recommendations are consistent with clinical practices and standards'. |
| 09 (Scientist) | SEND is user-friendly | This system is improving the quality of care | 'I understand how the system derived its recommendations, scores or alerts' and 'The system's recommendations are consistent with clinical practices and standards'. |
| 10 (Nurse band 5or 6) | SEND is user-friendly | I am able to provide better quality of care for patients by using this system | 'I understand how the system derived its recommendations, scores or alerts' and 'The system's recommendations are consistent with clinical practices and standard' |
| Selected statement | SEND is user-friendly | I am able to provide better quality of care for patients by using this system | Based on the participants' suggestions. This item is split in two:<br>1. I understand how the system derived its recommendations, scores or alerts.<br>2. The system's recommendations are consistent with clinical practices and standards |
| Justification | 'I like using SEND': participants might like using the system for many reasons: confusing statement. Participants were hesitating between the two statements: 'SEND is easy to use' or 'SEND is user friendly'. We decided to opt for the last statement as it was suggested by the participants. | The statement, 'This system is improving the quality of care', is too broad. Participants can't really know if the system is improving the quality of care, but they can judge the effect on their work. We think there is no real need to add 'I feel' at the beginning of the statement as suggested by two participants. | Pharmacists and physiotherapists were not interested to know about how the scores are derived but they don't mind knowing. Clinicians however were clear they will not use a system if they don't understand a minimum of how the score and recommendations are derived. The last few participants suggested to keep the original statement and to add the suggested statement as an independent item. |

test and the Kaiser-Meyer-Olkin (KMO) measure analyses. The KMO value varies between 0 and 1, with values closer to 0 indicating that factor analysis is inadequate. Exploratory factor analyses, on the other hand, have a high adequacy and can create distinct and reliable subscales or factors when the value is close to one.[73] KMO values between 0.5 and 07 are

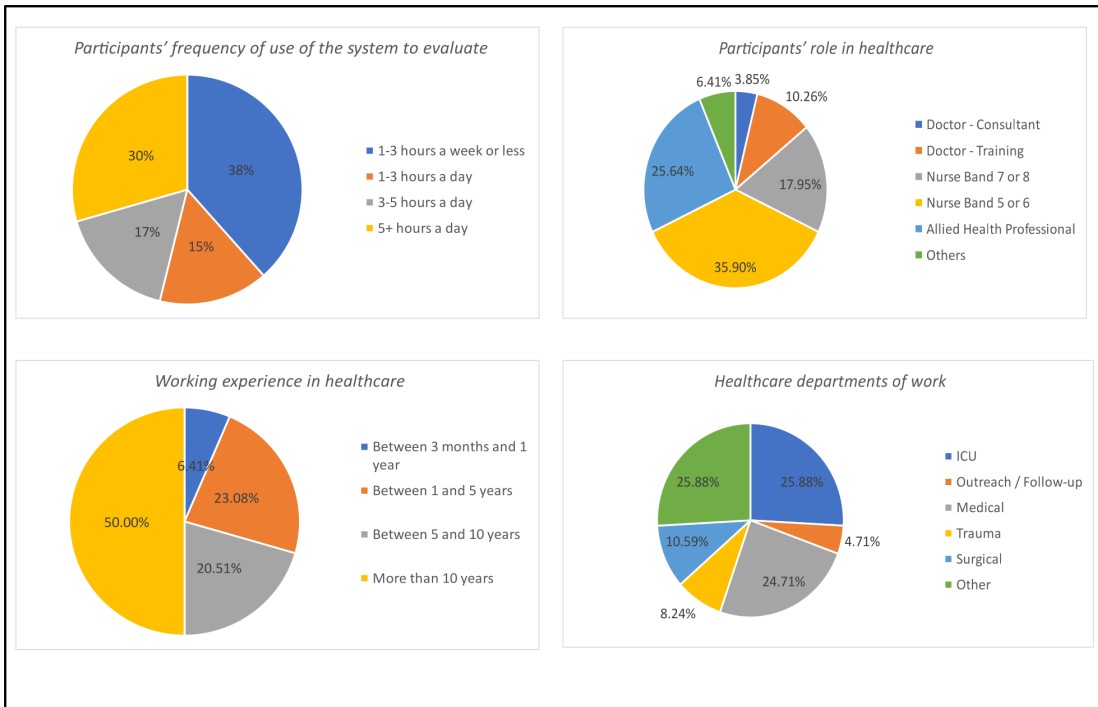

**Figure 3** Background information.
ICU = intensive care unit

considered mediocre by Kaiser; 0.7 and 0.8 as good, 0.8 and 0.9 as great, and >0.9 as superb.

To explore the factor structure of our new HSUS, we used the principal component analysis (PCA) with a varimax rotation as the extraction method due to its high reliability and lack of error.[69] The inclusion or exclusion of an item in a construct was determined by factor loadings >0.5, eigenvalues >1, the scree plot and the total variance are explained by the domain or factor.[74]

All 23 items of the HSUS were analysed using exploratory factor analysis. The resulting KMO was 0.818, indicating 'great' on the Kaiser scale. The Bartlett test was similarly positive with a p<0.001 (figure 5). As the KMO of the HSUS was above 0.8, and all the variables had loadings >0.5, the sample size is considered adequate for Factor Analysis and can be tested to see whether the scale performs as intended.

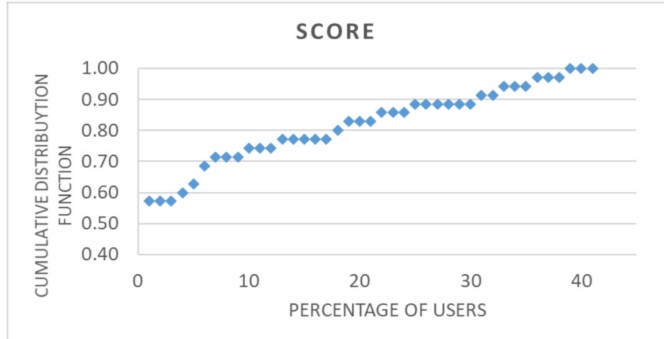

**Figure 4** Cumulative distribution function of the subscale 'patient safety and decision effectiveness' for send system.

When Kaiser's criterion and PCA were applied to the HSUS, five factors had eigenvalues ≥1.00. The exploratory factors analysis identified five domains or subconcepts which explained 68.62% of the data variance. Commonalities of the 23 items ranged from 0.515 to 0.867. Four of the factors had at least three items while the fifth only had one. For a better interpretation, we built the scree plot which indicated there were 3–5 factors which account for the variance (figure 6).

Only three statements were judged unclear once by two different participants ('It is easier to make efficient decisions by using (name of the system); (name of the system)'s recommendations are consistent with clinical practices and standards; (name of the system) helps improve patient outcomes'. Factor 5 ('A recommendation (false alert or score) is unlikely to impact patient safety') was judged twice as an unclear statement and as this item was covered by the rest of the scale, we decided to delete it to reassess the statistical analysis. This revised analysis shows the best statistically and conceptually appropriate solution for the new HSUS is a four-factor scale. The KMO of the final 22 item HSUS is 0.823 (figure 7).

When Kaiser's criterion and PCA were applied to the 22 items, four factors had eigenvalues ≥1.00 and the EFA results identified 4 factors or subconcepts where the commonalities of the 22 items ranged from 0.501 to 0.874. This helps to identify and localise the usability issues clearly and easily in each factor.

The first factor evaluates the utility perception of the system. For CDSS, this relates to patient safety and decision effectiveness. This factor is formed of seven items,

**Table 2**  Utility of HSUS open-ended questions: examples from send versus CERNER Millenium cohort

| CERNER | SEND |
|---|---|
| **I needed help with** | |
| ► Navigation, layout, more clinically designed (congruent with systems I am used to)<br>► Is not user friendly-too complex, too many different places to go. For example: tasks/interactive view/care plans- all need to be done every shift for every patient but all in different places and you need to search for them in a big list.<br>► Uploading photos correctly. navigating through the system—finding what relevant to my practice and how to successfully use. it can freeze or be slow. We often miss patients not referred. more information needed on occasion<br>► Scheduling appointments find notes find coding | ► Initial login, etc<br>► Nothing<br>► Nothing<br>► Nothing<br>► Nothing<br>► Nothing I am happy with using SEND<br>► Nothing currently<br>► Nothing<br>► Correcting data if error used<br>► Nothing |
| **I would add these functions or features** | |
| ► CERNER should have the 'continuous doc' feature added at OUH to make searching much easier<br>► Per shift: all the care plan+tasks + interactive view in the same place and then other things could be added as needed.<br>► instead of having different encounters on the same patient I would have the patient file divided by specialties. Add research function (specialties) | ► Total flow while on nasal high flow systems<br>► More detailed information regarding normal ranges for individuals. Some standard parameters do not apply to all individuals<br>► Link to labs<br>► Flow rate and FiO2 for high flow nasal oxygen<br>► Accurate recording of patients on HFNC oxygen<br>► Target SpO2 and adjust the traffic light system to reflect this (chronic Respiratory conditions vary so these need to be able to be adjusted.<br>► Flow rate with Airview<br>► Cap refill, Urine output - if applicable<br>► Oxygen therapy should add to the trigger score. Some patients can have a score of 0 but be on 8 L of oxygen therapy.<br>► If patient has specific SpO2 that is, COPD aim SpO2, to be reflective on SEND screen. Ability to cancel incorrect data after recorded.<br>► When recording blood pressure, there is an option for lying and standing, maybe add the option of sitting up right.<br>► Urine output and fluid balance all under one system to see trends along with the vital signs this may increase compliance. |
| **I would change the following** | |
| ► Layout, accessibility, observation recording, search function, documentation.<br>► Add on the available functionality such as continuous doc<br>► Not have different encounters -Add a search facility by using coding structure.<br>► Able to build theatre lists and upload TIMS data+Bluespier Add oncology data including chemo & radiotherapy | ► Better graphic representation<br>► Add in flow rate for high flow oxygen<br>► Increase the login timeout<br>► Make it more like HAVEN<br>► Consideration of link with ICCA for translation of date from one system to the other without duplication. Or the ability to go back further than 4 hours to input data as it is duplication of records.<br>► Change to NEWS2 regarding oxygen saturations |
| **Please use the space below to provide any other comments:** | |
| ► It is a poor, pretty unusable system.<br>► more training/update sessions to keep up with the changes. Logon can be temperamental! it slows work efficientcy<br>► Generally CERNER has been a fantastic tool to help me to complete my job role. I can quickly check the status of patients and prioritise them accordingly. In order to ensure more staff use the system for patient referrals and communication, it needs to be much more user friendly/ replicate the 'look/feel' of more established systems. This will ensure patient data is shared safely and securely<br>► When review of medications is requested there is a field to highlight not medically responsible. This generates lots of lists on power note until reviewed by medic but think this should be placed in a different format rather than producing reams of lists<br>► It feels like a very clunky system which is slow and takes a long while to achieve what is required<br>► It'is really time consumming, slow, not always making our time efficient seems to be all over the place, needs more structure to the components of nursing care that are needed<br>► The safety messages that pop up when you log in have the 'Don't show until new info posted' box but this doesn't work—I keep getting the same message for days or weeks despite using it. | ► Very useful<br>► Nothing to add<br>► I like how if you have starred your patients and discharged them they reappear on your page if readmitted to hospital.<br>► I feel SEND system takes away the clinical thinking and the judgement of staff to make clinical decisions<br>► It takes a bit while to learn and remember about various functions on EPR. Also I find it a bit difficult to correct errors of documentation |

COPD, Chronic Obstructive Pulmonary Disorder; EPR, Electronic Patient Record; FiO2, Fraction of inspired oxygen; HFNC, High Flow Nasal Cannula; HSUS, Healthcare Systems Usability Scale; ICCA, Intellispace Critical Care and Anaesthesia; NEWS/NEWS2, National Early Warning Score/National Early Warning Score 2; OUH, Oxford University Hospitals NHS Foundation Trust; SpO2, oxygen saturation; TIMS, Theatre Information Management System.

reflecting its importance in system adoption. Factor two includes six items concerned with the ease of use of the system which facilitates integration within the clinical workflow. Factor three with five items reflects the work effectiveness and cognitive workload and the final factor with four items relate to the user's perception of control. The final scale with the factor loadings of the final PCA are shown in table 3.

**KMO and Bartlett's Test**

| Kaiser-Meyer-Olkin Measure of Sampling Adequacy | | 0.818 |
|---|---|---|
| Bartlett's Test of Sphericity | Approx. Chi-Square | 1175.252 |
| | df | 253 |
| | Sig. | <0.001 |

**Figure 5** KMO and Bartelett's test of the 23 Items scale. KMO, Kaiser-Meyer-Olkin.

**KMO and Bartlett's Test**

| Kaiser-Meyer-Olkin Measure of Sampling Adequacy | | 0.823 |
|---|---|---|
| Bartlett's Test of Sphericity | Approx. Chi-Square | 1134.185 |
| | df | 231 |
| | Sig. | <0.001 |

**Figure 7** KMO and Bartlett's test of the 22 Items scale. KMO, Kaiser-Meyer-Olkin.

## Internal consistency and reliability

To conduct the reliability analysis, we calculated the Cronbach's alpha coefficient which is a measure of the internal consistency of the question scores. It depends on the interquestion correlations and the total number of questions.

Reliability contributes largely to the validity of a questionnaire as it refers to the extent to which the results obtained by the instrument can be reproduced. Internal consistency reflects the degree to which items on the scale or subscale are measuring same concept. Reliability can be established using a pilot test with 20–30 participants.[48] To study the inter-item correlations within the HSUS, we calculated the Cronbach's alpha coefficient,[75] which is a measure of the internal consistency of the item scores. Cronbach's alpha was calculated for each subscale as well as for the complete scale. Cronbach's alpha was 0.914 for the HSUS scale, indicating a high inter-items correlation, suggesting the scale is consistently reliable. The Cronbach's alpha of the four subscales also exceeded 0.7, which is the minimum suggested for a new instrument[62 76] (see table 3).

## DISCUSSION

Challenges in developing a usability scale in the clinical context are significant, and few guidelines exist in the literature. The specificity of the target user is the first challenge to face as health professionals are experiencing growing level of stress and cognitive workload. While new technologies may improve the quality of care and assist health professionals in their decision making, healthcare professionals will continue to reject any technology that they do not perceive its utility or that does not fit appropriately in their workflow.[6 30] Another challenge in the

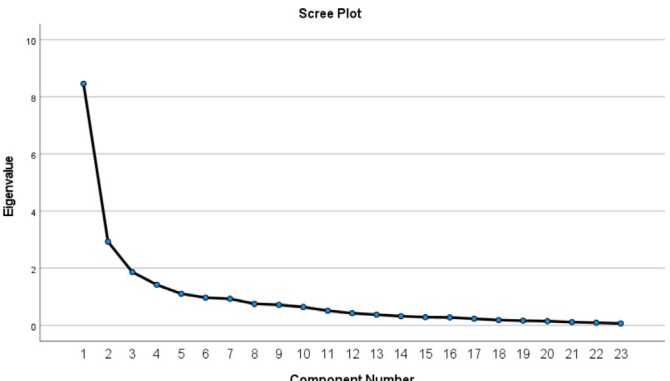

**Figure 6** Scree plot of the 23 items questionnaire.

healthcare environment is patient safety and quality of care, both of which are aspects associated with usability of health information and clinical decision-making systems. When a new healthcare system is established, it should improve patient outcomes while also ensuring the avoidance of errors and adverse effects on patients and/or the wider healthcare system itself. Another challenge is deciding whether these needs must be balanced or whether the necessity of effectiveness (safety, workflow) should be emphasised. This HSUS was specifically created with the intent of taking the context of use into account.

The review of existing CDSS systems, usability scales and previous research studies urged the need for a short, easy to use, reliable and accurate usability scale for HIS. To the best of our knowledge, this HSUS is the first validated scale for testing the usability of CDSS or HIS that also takes into account the specific clinical context. We have shown that the HSUS is an efficient method for collecting subjective opinions and experiences from a large number of potential users, as well as analysing usability issues. Dividing the HSUS into four subscales is a novel feature that allows evaluators to identify specific usability issues and investigate each subscale separately. The fact that is easy to complete, brief and can be delivered electronically makes it simpler to reach large numbers of users. The HSUS was created for health professionals with varying degrees of computer ability or literacy, while keeping in mind that they may be busy, distracted or interrupted.

This scale can provide critical information to address the most important usability challenges to the design and subsequent use of complex CDSS. In addition to the HSUS, we recommend that other usability testing methods, such as interviews and focus groups, are also required for a more thorough and realistic evaluation, as well as to evaluate whether the participants' behaviours match well with the results from the HSUS. The HSUS will detect early usability issues, identifying the source of problems, and pinpointing the threat rate. It is difficult to assess a system after the initial implementation. Usability scales that are used after a quick review of a new system during a usability test session tend to be less realistic than tools used after a few days or a short-term trial. It may be difficult and inaccurate for users to evaluate systems after their first interaction. It is preferable to use the usability scale a few days after implementation to allow some degree of familiarity to form. This valuable and novel work was developed and validated through several phases

**Table 3** Final HSUS with factor loadings and Cronbach's alpha

| Items | Utility: Patient safety and decision effectiveness Factor 1 | Ease of use: Workflow Integration Factor 2 | Work Effectiveness Factor 3 | User Control Factor 4 |
|---|---|---|---|---|
| (System name) helps me work more efficiently | 0.734 | | | |
| (System name) makes it easier to collaborate with colleagues | 0.758 | | | |
| I am able to provide better quality of care for patients by using (system name) | 0.613 | | | |
| It is easier to make efficient decisions by using (system name) | 0.706 | | | |
| (System name) helps improve patient outcomes | 0.642 | | | |
| (System name) helps prevent clinical errors | 0.677 | | | |
| (System name) generates a useful summary view of the patient's current health status | 0.503 | | | |
| (System name) fits well with the way I currently work | | 0.609 | | |
| I found the information provided on the screen understandable | | 0.818 | | |
| I found it easy to navigate through (system name) | | 0.847 | | |
| I can easily remember how to use (system name) | | 0.801 | | |
| The screen layout makes it easy to see each piece of information | | 0.84 | | |
| On the screen, I can find specific information I need quickly | | 0.803 | | |
| (System name) helps me prioritise my daily workload | | | 0.675 | |
| I understand how (system name) creates its recommendations (scores or alerts) | | | 0.67 | |
| (System name) 's recommendations (scores or alerts) are consistent with clinical practices and standards | | | 0.704 | |
| (System name) includes all the relevant information I need | | | 0.678 | |
| I believe the recommendations (scores or alerts) are reliable | | | 0.758 | |
| (System name) supports my decision making rather than dictating it | | | | 0.516 |
| It is easy to correct a data entry error in (system name) | | | | 0.758 |
| (System name) highlights potential data entry errors | | | | 0.781 |
| Recommendations (or alerts) do not unnecessarily interrupt my workflow | | | | 0.506 |
| Cronbach's alpha coefficient | 0.873 | 0.926 | 0.777 | 0.702 |

HSUS, Healthcare Systems Usability Scale.

## Scale development

The HSUS was developed using a rigorous, iterative and user-centred approach that consisted of face validity, content validity, exploratory factor analysis and reliability testing. First, we proposed items based on our experience from previous evaluation of the HAVEN system. We complimented this knowledge with a literature review and discussions with various clinicians working with the research team. A panel of seven usability experts in forms and surveys, plain language, user experience and health content designers contributed to the content validity. The number of experts for content validity was sufficient to agree consensus on the main concept.[59] Three of these experts recommended another review of the revised version at the end of the HSUS usability testing or face validity phase.

Face validity of the HSUS was established by an iterative review of the questions by 10 health professionals recruited for a semistructured interview. Each participant's suggestions were incorporated into the following iteration for evaluation, resulting in a total of five iterations of the scale. Face validity was established when we noticed agreement among the last four participants. The participant variability and concept saturation determined the sample size for the qualitative work.[77 78]

During the content and face validity phases of the HSUS, items were deleted, changed and new items were suggested to ensure that all system usability aspects were addressed, and the questionnaire items were well designed and written, with no common errors such as double negative, confusing, or double-barrelled items. The number of items in the HSUS ranged from 21 at the outset to 28 for face validity and finally, after statistical evaluation, to 22. The exploratory factor analysis defined four factors: patient safety and decision effectiveness, workflow integration, work effectiveness, user control. These factors covered issues related to the patient safety, the ease of use of the system, task's complexity, the alerts or recommendations utility, the effects of the system on the workflow, the quality of care and ability of the clinician to take an informed and timely decision. The HSUS presented a very good reliability evaluation with a global Cronbach's alpha of 0.914 and subscale alphas ranging from 0.702 and 0.926.

While the open-ended questions were optional, they had several advantages when used within the HSUS questionnaire. They allowed participants to independently and creatively comment using their own lexicon without being influenced by the interviewer and it revealed important issues to investigate further. Free-text responses can also provide quick feedback that may reveal unexpected findings.

## HSUS subscales

### Patient safety and decision effectiveness

CDSS should, in theory, make daily tasks simpler, resulting in increased efficiency and safety. If the performance of health professionals is compromised, work efficiency is reduced, and the quality of care and patient safety may be impacted. A lack of usability in this subscale should be concerning, as the usage of this system could compromise patient safety. This part reflects the perceived utility by the health professionals.

### Workflow integration

This subscale is more related to the system ease of use while focusing on the health professionals' concerns. In a busy environment, clinicians need a system that fit well in their workflow, understandable and effective in a timely and accurate way.

### Work effectiveness

When offering clear, logical and practical guidance (warning or score) in a way that fit well into the clinical workflow, CDSS has the potential to be extremely useful and improve health professionals work effectiveness.

### User control

Health professionals must feel in control of the any new technology system which needs to be responsive to any internal or external change.

## Using the HSUS; customisation for real-world implementation

The online survey was designed to take 10–15 min to complete due to the time constraints of the health professionals. Some items may not be applicable to any CDSS or HIS. These items can be omitted. The options 'unclear statement' and 'not applicable' were added for validation purposes. We needed to know if any statements in the HSUS were unclear. For future use of the HSUS, these two options may be omitted, and the final scale will consist of 22 items rated on 5-point Likert scales.

We recommend tailoring the questionnaire to each application and naming the system within the question (replace the word 'system'). Depending on the system's functions, the questionnaire should be customised to use the terms 'recommendations,' 'scores,' or 'alerts' (table 3).

The complementary information gathered from the background questionnaire could be useful for HSUS data analysis. The background questionnaire should be kept short and include only those questions that are directly relevant to the needs of the study.

The HSUS can be used for the assessment of the severity of usability violation of each subscale. Each item is rated on a 1–5 scale; 1=strongly disagree (crucial usability concern), 2=disagree (major usability concern), 3=neutral (minor usability concern), 4=agree (usability can be improved) and 5=strongly agree (no usability concerns). A higher score reflects a higher level of agreement on each item. We calculated the overall group average for each item followed by the average for each subscale. The higher the average score for each subscale, the higher the level of usability perceived by participants. The overall score of this scale is the average of the four subscale scores. A usability score between 20% and 50% indicates a critical need to address the system's usability

issues; between 50% and 70% indicates a need to address the system's usability concerns, some of which may be major; between 70% and 90% indicates a good usability score with the potential to improve; and between 90% and 100% indicates an excellent and easy to use system. The researcher or investigator must decide if particular requirements, such as patient safety and decision effectiveness, should be prioritised.

Decision effectiveness and patient safety were combined statistically and logically due to the relationship of efficiency to safety.[30 79] If health professional performance is affected, work efficiency is slowed and quality of care and patient safety may be jeopardised. A lack of usability in this subscale should raise alarms as the use of the system under evaluation might endanger patient safety.

## LIMITATIONS

A possible limitation is the relatively small number of participants even though some experts recommend a sample size of 12–50 for a pilot study.[80–82] While the KMO test value revealed that the sample size was sufficient for factor analysis, the findings of the factor analyses may differ with greater number of participants. To strengthen the consistency of the scale and support its standardisation, further work could be undertaken in a larger sample.

This study was limited to healthcare professionals based in the secondary care environment. When used under different contexts or working circumstances, validity testing may be required, but the HSUS could be applied to other types of clinical system in other healthcare settings.

## CONCLUSION

In this paper, we have presented novel work to develop a validated and reliable instrument to measure the usability of electronic systems in the healthcare environment. The HSUS can be useful in research institutes and hospitals to improve the usability of a newly designed or implemented CDSS or HIS in an efficient way. When possible, HSUS should be administered a few days following the implementation of the new system to allow for a period of familiarisation. Because it has been designed for self-administration and brevity, this scale offers an easy way to assess the usability of clinical systems that can be used alongside other techniques. This offers a better understanding of the usability issues in context and allows designers to address these before potential medical adverse events can occur.

The HSUS is divided into four subscales, namely patient safety and decision effectiveness, workflow integration, work effectiveness and user control. The free-text comments enhance this scale's potential to provide insight into what is in the health professional mindset. As part of the overall analysis, these constructs are useful as they allow to compare systems based on their subscale scores. By determining which construct is the root of the problem, usability concerns may be more easily localised.

**Acknowledgements** We would like to express our sincere acknowledgements to the international panel of experts who evaluated the scale and the usability testing from their homes and institutions in the United States of America (Maryland, Missouri, Pittsburgh, and Utah) and the UK (Manchester, Milton Keynes and Oxford); Thank you for all the questionnaire respondents as well as the HAVEN research team members who helped in developing and evaluating the HSUS.

**Contributors** AG was in charge of the study design, data collection, evaluation, analysis and paper writing. JLD helped with the study design, paper drafting and proofreading. MWW and PJW contributed to the study design and approved the final manuscript. PJW is the chief investigator of the project and acts as guarantor for this work.

**Funding** This work was supported by the Health Innovation Challenge Fund grants HICF-R9-524 and WT-103703/Z/14/Z; a parallel funding partnership between the UK Department of Health and Social Care and the Wellcome Trust.

**Disclaimer** The views expressed in this publication are those of the authors and not necessarily those of the UK Department of Health and Social Care or the Wellcome Trust.

**Competing interests** None declared.

**Patient and public involvement** Patients and/or the public were involved in the design, or conduct, or reporting, or dissemination plans of this research. Refer to the Methods section for further details.

**Patient consent for publication** Not applicable.

**Ethics approval** Ethics approval was granted by South Central Oxford C Research Ethics Committee on 13 June 2016 and the REC reference is 16/SC/0264. Participants gave informed consent to participate in the study before taking part.

**Provenance and peer review** Not commissioned; externally peer reviewed.

**Data availability statement** Data are available on reasonable request. Data may be available on reasonable request from the chief investigator.

**ORCID iDs**
Abir Ghorayeb http://orcid.org/0000-0003-0207-580X
Julie L Darbyshire http://orcid.org/0000-0002-7655-1963
Marta W Wronikowska http://orcid.org/0000-0002-5416-7872
Peter J Watkinson http://orcid.org/0000-0003-1023-3927

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
