## [Reviewer comments · BMJ Open]

ARTICLE DETAILS

TITLE (PROVISIONAL)	Design and validation of a new Healthcare Systems Usability Scale (HSUS) for Clinical Decision Support Systems: a mixed methods approach
AUTHORS	Ghorayeb, Abir; Darbyshire, Julie; Wronikowska, Marta Weronika; Watkinson, Peter

VERSION 1 – REVIEW

REVIEWER	John, Denny Amrita Institute of Medical Sciences and Research Centre
REVIEW RETURNED	20-Jul-2022

GENERAL COMMENTS	The authors have done a good study however the mixed methods study design is not detailed as per mixed method study designs. The authors are suggested to look into https://www.ncbi.nlm.nih.gov/pmc/articles/PMC5602001/ and make appropriate modifications in the methods section including reporting of results. Along with this the authors are also suggested to use the MMAT tool to check the completeness and quality of the final manuscript https://www.pcne.org/upload/wc2013/Lectures/Pluye.pdf
---

REVIEWER	Kharroubi, Samer American University of Beirut, Department of Nutrition and Food Sciences
REVIEW RETURNED	31-Oct-2022

GENERAL COMMENTS	Thank you for inviting me to review this manuscript. In brief, the paper presents the design and validation of a new Healthcare Systems Usability Scale using mixed research methods. It describes some very interesting findings that can be very useful for early identification of usability issues that may impact patient safety and quality of care. There is a critical need for usability scale that takes into consideration different usability aspects in medical context which supports the importance of this work. My impression is that the method proposed here may have scientific potential. I also believe that the subject may matter of relevance to BMJ Open journal. Below, a list of minor modifications to be addressed by the co-authors: 1. A good rationale and background is provided. Most recent studies on the usability of healthcare systems can be cited like:
---

	• Poncette, Akira-Sebastian, et al. "A Remote Patient-Monitoring System for Intensive Care Medicine: Mixed Methods Human-Centered Design and Usability Evaluation." JMIR Human Factors 9.1 (2022): e30655. • Hajesmaeel-Gohari, S., Khordastan, F., Fatehi, F., Samzadeh, H., & Bahaadinbeigy, K. (2022). The most used questionnaires for evaluating satisfaction, usability, acceptance, and quality outcomes of mobile health. BMC Medical Informatics and Decision Making, 22(1), 1-9. 2. In the abstract, Page 2, line 19: The sentence is not clear, maybe there should be a comma after "patient safety" instead of a colon. Please amend. 3. Page 7, line 50: It is better to replace this sentence at the end of the paragraph (line 21, page 8) given the authors speak about the aim of the paper twice. 4. Page 22, first paragraph: The authors claim that the scale offers three open-ended questions while Table 2, which offers illustrations of the open-ended questions of HSUS, presents 4 open-ended questions! This needs clarification. 5. Page 24, line 11: The reference number 60, is it for how the inclusion of an item in a construct is determined or it is just related to factor loadings? The rest of the claims in this paragraph on this page should be supported by references. 6. The authors should start the discussion by emphasizing on the importance of HSUS as there is need for a reliable and accurate usability scale for health information systems. This would make the discussion stronger. I suggest starting the discussion by line 9 page 28 to line 44, page 29. Then the authors can talk about the methodology or the validation process of the scale. 7. In the conclusion, line 52: Is HSUS designed to evaluate just the newly designed systems? In the discussion, the authors have suggested to evaluate the systems after certain time of use. 8. The authors may perhaps want to summarise the different subscales in the discussion 9. Minor points to be addressed:  Line 31 needs editing. The justification line of Table 1 needs editing. Page 11, line 40: Missing full stop. Page 31, line 17: Reference link error Page 12, line 58: Figure link error Page 27, line 25: Space between coefficient and the reference number (62). Page 27, line 37: Table reference error Overall, I found the findings very interesting and I think this is a nice piece of work to inform further technological development in this area.
--	---

VERSION 1 – AUTHOR RESPONSE

Responses to reviewer 1 comments:

1. It describes some very interesting findings that can be very useful for early identification of usability issues that may impact patient safety and quality of care. There is a critical need for usability scale that takes into consideration different usability aspects in medical context which supports the importance of this work. My impression is that the method proposed here may have scientific potential. I also believe that the subject may matter of relevance to BMJ Open journal.

Thank you, we are glad that this reviewer found that our study provided interesting findings that can inform smart home development. We had had hoped that our paper would provide this to readers and we are pleased.

2. A good rationale and background is provided. Most recent studies on the usability of healthcare systems can be cited like:

- Poncette, Akira-Sebastian, et al. "A Remote Patient-Monitoring System for Intensive Care Medicine: Mixed Methods Human-Centered Design and Usability Evaluation." *JMIR Human Factors* 9.1 (2022): e30655.

- Hajesmaeel-Gohari, S., Khordastan, F., Fatehi, F., Samzadeh, H., & Bahaadinbeigy, K. (2022). The most used questionnaires for evaluating satisfaction, usability, acceptance, and quality outcomes of mobile health. *BMC Medical Informatics and Decision Making*, 22(1), 1-9.

Thank you, this is an interesting suggestion. More details are added in the "Usability" and "Need for a new usability scale for clinical interfaces" sections (Page 5 and 6).

3. In the abstract, Page 2, line 19: The sentence is not clear, maybe there should be a comma after "patient safety" instead of a colon. Please amend.

Thank you. This has been corrected as per your suggestion.

4. Page 7, line 50: It is better to replace this sentence at the end of the paragraph (line 21, page 8) given the authors speak about the aim of the paper twice.

Thank you. This has been corrected as per your suggestion.

5. Page 22, first paragraph: The authors claim that the scale offers three open-ended questions while Table 2, which offers illustrations of the open-ended questions of HSUS, presents 4 open-ended questions! This needs clarification.

Thank you and we are sorry for this typographical error. There were four open ended questions as described in Table 2. We have made appropriate correction to the text of the revised version, "Face validity" section (Page 11) and "Content validity" section (Page 15).

6. Page 24, line 11: The reference number 60, is it for how the inclusion of an item in a construct is determined or it is just related to factor loadings? The rest of the claims in this paragraph on this page should be supported by references.

Thank you. For more clarity, the reference was replaced at the end of the sentence.

7. The authors should start the discussion by emphasizing on the importance of HSUS as there is need for a reliable and accurate usability scale for health information systems. This would make the discussion stronger. I suggest starting the discussion by line 9 page 29 to line 44, page 30. Then the authors can talk about the methodology or the validation process of the scale.

Thank you for this suggestion, the discussion was updated as suggested.

8. In the conclusion, line 52: Is HSUS designed to evaluate just the newly designed systems? In the discussion, the authors have suggested to evaluate the systems after certain time of use.

Thank you, a clarification was added to the conclusion, page 34.

9. The authors may perhaps want to summarise the different subscales in the discussion

Thank you for this suggestion, HSUS subscales summary was added on page 31.

10. Minor points to be addressed:

- a. Line 31 needs editing.
- b. The justification line of Table 1 needs editing.
- c. Page 11, line 40: Missing full stop.
- d. Page 31, line 17: Reference link error
- e. Page 12, line 58: Figure link error
- f. Page 27, line 25: Space between coefficient and the reference number (62).
- g. Page 27, line 37: Table reference error

Thank you and we are sorry about these errors. We have revised the text and made the edits in light of the suggestions. We are grateful for the attention to detail that this reviewer has provided.

11. Overall, I found the findings very interesting and I think this is a nice piece of work to inform further technological development in this area.

Thank you, we are glad that this reviewer found that our study provided interesting findings that can inform further technological development. We had had hoped that our paper would provide this to readers and we are pleased.

Responses to reviewer 2 comments:

1. The authors have done a good study however the mixed methods study design is not detailed as per mixed method study designs. The authors are suggested to look into <https://www.ncbi.nlm.nih.gov/pmc/articles/PMC5602001/> and make appropriate modifications in the methods section including reporting of results.

Thank you, this is an interesting suggestion. While we read carefully the paper entitled “ How to Construct a Mixed Methods Research Design” [1], we believe we followed appropriate methods as described in [2-8] . . To develop and validate a new scale to evaluate the usability of any electronic healthcare system we followed the methodology as described in [2-8] as well as on the Imperial College London website (<https://www.imperial.ac.uk/education-research/evaluation/tools-and-resources-for-evaluation/questionnaires/how-a-scale-is-developed/>).

2. This methodology has been used in the validation of different scales [8-13]. To make our process clearer, we added a few lines on the first paragraph of the "Methods" section, Page 9. Along with this the authors are also suggested to use the MMAT tool to check the completeness and quality of the final manuscript <https://www.pcne.org/upload/wc2013/Lectures/Pluye.pdf>

Thank you for this suggestion. Through this paper we present a combination of quantitative and qualitative methods. The MMAT was designed for systematic mixed studies reviews [14]. We do recognise though that the MMAT can also be used to appraise the quality of empirical studies. Under the MMAT guidelines, our work is an exploratory design (QUALITATIVE proposal, followed by QUANTITATIVE generalization/tool development). The results are concluded by sequential synthesis: we analysed the qualitative study before the quantitative. We have added this clarification to the methods section.

References

1. Schoonenboom, J. and R.B. Johnson, How to construct a mixed methods research design. *KZfSS Kölner Zeitschrift für Soziologie und Sozialpsychologie*, 2017. 69(2): p. 107-131.
2. Yaddanapudi, S. and L. Yaddanapudi, How to design a questionnaire. *Indian journal of anaesthesia*, 2019. 63(5): p. 335.
3. DeVellis, R.F. and C.T. Thorpe, *Scale development: Theory and applications*. 2021: Sage publications.
4. Boateng, G.O., et al., Best practices for developing and validating scales for health, social, and behavioral research: a primer. *Frontiers in public health*, 2018. 6: p. 149.
5. Artino, A.R., Jr., et al., Developing questionnaires for educational research: AMEE Guide No. 87. *Med Teach*, 2014. 36(6): p. 463-74.

6. Bolarinwa, O.A., Principles and methods of validity and reliability testing of questionnaires used in social and health science researches. Nigerian Postgraduate Medical Journal, 2015. 22(4): p. 195.
7. Collingridge, D., Validating a questionnaire. Retrieved Febr, 2014. 25: p. 2017.
8. Parsian, N. and T. Dunning, Developing and validating a questionnaire to measure spirituality: A psychometric process. Global journal of health science, 2009. 1(1): p. 2-11.
9. Tsang, S., C.F. Royse, and A.S. Terkawi, Guidelines for developing, translating, and validating a questionnaire in perioperative and pain medicine. Saudi journal of anaesthesia, 2017. 11(5): p. 80.
10. Bai, Y., et al., Development and validation of a questionnaire to evaluate the factors influencing training transfer among nursing professionals. BMC Health Services Research, 2018. 18(1): p. 1-9.
11. Desalu, O.O., et al., Development and validation of a questionnaire to assess the doctors and nurses knowledge of acute oxygen therapy. PLoS One, 2019. 14(2): p. e0211198.
12. Kaitelidou, D., et al., Development and validation of measurement tools for user experience evaluation surveys in the public primary healthcare facilities in Greece: a mixed methods study. BMC family practice, 2019. 20(1): p. 1-12.
13. Schnall, R., H. Cho, and J. Liu, Health Information Technology Usability Evaluation Scale (Health-ITUES) for usability assessment of mobile health technology: validation study. JMIR mHealth and uHealth, 2018. 6(1): p. e8851.
14. Hong, Q.N., et al., Mixed methods appraisal tool (MMAT), version 2018. Registration of copyright, 2018. 1148552(10).

VERSION 2 – REVIEW

REVIEWER	John, Denny Amrita Institute of Medical Sciences and Research Centre
REVIEW RETURNED	12-Dec-2022

GENERAL COMMENTS	The manuscript is accepted in the current form.
---

REVIEWER	Kharroubi, Samer American University of Beirut, Department of Nutrition and Food Sciences
REVIEW RETURNED	19-Dec-2022

GENERAL COMMENTS	Many thanks for the revision. The authors have successfully managed to address all comments!
--